# Optimized Design of an Ultrasonic-Based High-Efficiency Wireless Passive Monitoring System for Sealed Metal Compartments

**DOI:** 10.3390/mi15010048

**Published:** 2023-12-26

**Authors:** Bowen Qi, Juan Cui, Yongqiu Zheng, Bingrui Zhang, Chengqun Chu, Xiaolong Yan, Xiang Gao, Chenyang Xue

**Affiliations:** 1The Key Laboratory of Instrumentation Science & Dynamic Measurement Ministry of Education, North University of China, Taiyuan 030051, China; qbw769844210@163.com (B.Q.); zhengyongqiu@nuc.edu.cn (Y.Z.); zbrchosenone@163.com (B.Z.); chuchengqun@nuc.edu.cn (C.C.); yanxl@nuc.edu.cn (X.Y.); 18734901562@163.com (X.G.); xuechenyang@nuc.edu.cn (C.X.); 2Department of mechanics, Jinzhong University, Jinzhong 030619, China

**Keywords:** ultrasonic, condition monitoring, high efficiency, sealed metal compartments (SMCs)

## Abstract

The condition monitoring (CM) of sealed metal compartments (SMCs) is an urgently required restructure. Ultrasound penetrates SMCs to power and communicate with built-in sensors, enabling the CM of SMCs. However, current ultrasonic wireless power transfer and data communication (UWPTADC) systems are large and complex, and limited by the efficiency of energy transfer and data reliability. In this paper, an optimized design of a high-efficiency wireless passive monitoring system using UWPTADC techniques is proposed for SMC. The circuit model of the system is developed and analyzed to achieve an optimal design for efficient wireless power transfer and effective data communication coupling. A test system was constructed using a steel wall of 11 mm thickness as a validation object. At the ultrasonic carrier frequency of 1.045 MHz, the system has an energy transfer efficiency of 60%, and a communication rate of 50 kbps. In addition, the system realizes temperature and humidity monitoring inside a 13 mm thick cylindrical SMC, simulating the process of ultrasonic CM of an actual engine compartment. The system provides a wiring-free and battery-free solution for CM in SMCs, advancing CM in aerospace, marine and other fields.

## 1. Introduction

Condition monitoring (CM) is a predictive maintenance approach that relies on the sensor network to monitor the operational status of equipment and detect faults and anomalies [1,2,3,4]. This method can effectively reduce equipment maintenance costs, prevent safety accidents, and extend equipment life. Existing sensors are mostly battery-powered or wired. Battery power needs to be replaced or recharged frequently, which is difficult to achieve in some sealed environments (engine compartments and ship cabins). Wired power requires drilling holes in the enclosure, which destroys the strength of the original structure and affects its sealing properties. To solve the above problems, wireless power transfer and data communication (WPTADC) technologies have been proposed [5,6,7]. Electromagnetic WPTADC technology is subject to electromagnetic shielding effects and cannot be used in the sealed metal compartment (SMC) [8,9]. However, power and data can be transferred efficiently by using piezoelectric transducers to send and receive ultrasound through the metal [10,11,12,13]. Many scholars have performed related research, but only the energy transfer or data communication is realized [14,15,16,17,18,19,20].

In order to realize both ultrasonic wireless power transfer and data communication (UWPTADC) in one system, Lawry et al. [21] proposed an UWPTADC system with dual electric–ultrasonic–electric (EUE) channels. Since the power transfer channel is separate from the data communication channel, this makes the system more complex and bulkier. Shoudy et al. [22] proposed a single channel for the UWPTADC system. The single EUE channel systems are significantly smaller and less complex than dual EUE channel systems. However, there are still problems such as low coupling between data communication and energy transfer between UWPTADC and high requirements for demodulation circuits. To solve the problem of low coupling between data communication and energy transfer between UWPTADC in a single EUE channel, our group proposed an optimized design approach [23]. Our solution leverages conjugate impedance matching and impedance modulation techniques to facilitate efficient energy transfer and data communication. Despite this progress, the current system is limited to a communication rate of 20 kbps and an energy transfer efficiency of 45.7%. Further improvements are needed to increase the performance of the system in these areas and to enable specific applications.

This paper presents a wireless passive and efficient CM system that uses ultrasound to power and communicate with embedded sensors inside SMC, avoiding wiring and battery replacement. The equivalent circuit model of the system is established and the design process of simultaneous conjugate impedance matching is determined. Based on the circuit model, the effects of different modulation resistors on the energy transfer efficiency and data communication quality are analyzed, and coupling for efficient energy transfer and effective data communication are achieved. Moreover, the system is highly integrated. The receiver and sensor module, mounted inside SMC, have an overall size of only 20 cm³. The external excitation and demodulation model powers SMC internally while demodulating the returned temperature and humidity data. The system has a promising future in a variety of industrial and military applications.

## 2. Overview of System

The block diagram of the ultrasonic CM system for SMC is shown in Figure 1. The system consists of three main parts: the excitation and demodulation module, the matched EUE (MEUE) channel, and the receiver and sensor module. The excitation and demodulation module consists of a master control unit (MCU), a waveform generation circuit (AD9833), and a signal demodulation circuit, while the MEUE channel consists of an input piezoelectric ceramic (IPZT), an output piezoelectric ceramic (OPZT), a coupler (K-9741), an SMC, and an impedance matching network. The receiver and sensor module consists of MCU, a rectifier regulator circuit, and temperature and humidity sensors (HIH8131) and impedance modulator.

IPZT and OPZT models H8P301000 and H5P301000, respectively, are manufactured by Electronic Ceramics Ltd. (Muang Lamphun, Thailand) The IPZT and OPZT are circular sheets, 30 mm in diameter and 2 mm thick, with a resonant frequency of 1 MHz; the direction of polarization is along the thickness, and the electrodes are flip-flopped. The excitation voltage applied to the IPZT causes it to vibrate in the thickness direction, generating an ultrasonic signal. To maximize the capture of the signal generated by the IPZT, the OPZT is coaxially mounted on the flat structure of SMC [11]. When ultrasound propagates from one medium to another, a larger difference in acoustic impedance results in a more pronounced reflection and a higher energy transfer loss. Given the large acoustic impedance difference between existing coupler materials (e.g., gels, silicones, and resins) and piezoelectric ceramics and metals, it is desirable to minimize the thickness of the coupler to reduce ultrasonic attenuation and reflection within it. In this paper, K-9741 epoxy resin is used as a coupling agent to create a strong bond between the IPZT, OPZT and SMC to ensure an effective acoustic connection.

The system enables wireless energy transfer to SMC. The signal generated by the AD9833 chip in the external excitation and demodulation module is amplified by a power amplifier to become a carrier signal driving the MEUE channel. To reduce unnecessary electrical reflections during energy transfer, an impedance matching network is connected to both ends of the MEUE channel to achieve simultaneous conjugate impedance matching between the two ports. The ultrasound generated by the IPZT penetrates SMC and is converted to AC by the OPZT. The AC voltage output from the MEUE channel is converted to a stable DC power by the receiver and sensor module.

The system enables data communication from inside to outside. Due to the strong coupling at both ends of the MEUE channel, the change in the equivalent impedance of the receiver and sensor module, realized by means of an impedance modulator, directly affects the input impedance of the MEUE channel. According to the voltage division principle, a change in the input impedance leads to a change in the amplitude of the input carrier signal of the MEUE channel, i.e., amplitude modulation is realized. The receiver and sensor module converts the monitored temperature and humidity data into a binary modulated signal and achieves amplitude modulation of the carrier signal by impedance modulation [24]. The excitation and demodulation module is responsible for demodulating the modulated carrier signal. The demodulation process includes envelope detection, filtering, amplification, voltage comparison, sampling, and judgement. Through the above energy transfer and data communication processes, the system enables the real-time monitoring of the temperature and humidity environment inside the sealed metal container.

## 3. System Equivalent Circuit Model Design

The equivalent circuit model of the system is shown in Figure 2 and is modeled as follows:

(1) EUE channel: A network analyzer is used to perform a scattering parameter (*S*-parameter) scan of the constructed EUE channel. According to Equation (1), the *Z*-parameter model of the EUE channel at different frequencies can be determined.

(2) Excitation and demodulation module: According to Thevenin’s theorem, a linear resistive single-port network with an independent power supply can be equivalently represented by a series circuit consisting of a voltage source and a resistor [25]. This representation accurately captures the port characteristics of the network. Therefore, we can model the excitation and demodulation module as a series circuit comprising a voltage source (*U_S_*) operating at frequency (*f*) and a resistor (*R_S_*).

(3) Receiver and sensor module: The receiver and sensor module is considered as a load terminal, represented by the resistor *R_L_*. Inside the module, there is an impedance modulation circuit comprising two MOSFETs connected back-to-back and a modulating resistor *R_M_*. These MOSFETs control the parallel connection and disconnection of the modulating resistor *R_M_* to the load resistor *R_L_*.

(4) Impedance matching network: The design process of the impedance matching network is explained in Section 3.1, and it aims to optimize the matching between the source and load impedances. Different circuit models can be designed for various parameter conditions.

The input impedance of the EUE channel is *Z_IN_* and the output impedance is *Z_OUT_*. The EUE channel is cascaded with the impedance matching networks of the input and output ports to form the MEUE channel, where the input impedance, output impedance, input voltage, and output voltage are *ZN_IN_*, *ZN_OUT_*, *V*_1_, and *V*_2_, respectively.

When the internal parameters of the system, such as the specifications of the piezoelectric ceramic, the thickness of the coupling layer, or the shape of SMC, are modified, the *Z*-parameters of the system change, while the equivalent circuit model remains unaffected. To recalibrate the *Z*-parameter values of the EUE channel, a new S-parameter scan can be performed using a vector network analyzer. Based on the equivalent circuit model, a suitable matching element can be selected for impedance matching and a suitable modulation resistor can be selected for amplitude modulation to enhance the energy transfer efficiency and data communication quality of the system.

### 3.1. Simultaneous Conjugate Impedance Matching

To reduce the electrical losses caused by impedance mismatch during energy transfer in the EUE channel, impedance matching is performed at the input and output of the EUE channel. Rahola first proposed a two-port simultaneous conjugate impedance matching technique [26]. The process of simultaneous conjugate impedance matching of the EUE channel is divided into three steps.

In the first step, the EUE channel is scanned using a network analyzer (ROHDE & SCHWARZ ZNB20) to obtain the *S*-parameters of the EUE channel at different frequencies. The *S*-parameters are converted to Z-parameters according to Equation (1) [27]. *Z*_0_ is the characteristic impedance of the ROHDE & SCHWARZ ZNB20.
(1)$$\left\{ \begin{array}{l} Z_{11}=Z_{0}\frac{(1+S_{11})(1-S_{22})+S_{12}S_{21}}{(1-S_{11})(1-S_{22})-S_{12}S_{21}}\\ Z_{12}=Z_{0}\frac{2S_{21}}{(1-S_{11})(1-S_{22})-S_{12}S_{21}}\\ Z_{21}=Z_{0}\frac{2S_{12}}{(1-S_{11})(1-S_{22})-S_{12}S_{21}}\\ Z_{22}=Z_{0}\frac{(1-S_{11})(1+S_{22})+S_{12}S_{21}}{(1-S_{11})(1-S_{22})-S_{12}S_{21}} \end{array}\right.$$

The second step is to substitute the *Z*-parameters of the EUE channel at different frequencies into Equations (2) and (3) to find the input impedance *Z_IN_* and the output impedance *Z_OUT_* of the EUE channel when simultaneous conjugate impedance matching is performed, where Re{}, Im{} and * denote the real part, the imaginary part and the conjugate of the parameters, respectively. It should be noted that the appropriate symbols should be chosen to ensure that the real parts of *Z_IN_* and *Z_OUT_* are positive.
(2){ZIN*=α1±Δ2Re{Z22}ZOUT*=α2±Δ2Re{Z11}
(3){α1=−2jRe{Z22}Im{Z11}+jIm{Z12Z21}α2=−2jRe{Z11}Im{Z22}+jIm{Z12Z21}Δ=(2Re{Z11}Re{Z22}−Re{Z12Z21})2−|Z12Z21|2

Finally, Since the *Z_IN_* and *Z_OUT_* of the EUE channel in the simultaneous conjugate impedance matching state obtained in the second step vary with *f*, the corresponding impedance matching network is designed according to the relationship between *Z_IN_* and *R_S_* and the relationship between *Z_OUT_* and *R_L_*. The eight topologies shown in Figure 3 are specific circuit models of the impedance matching network in Figure 2. *R_IN_*, *X_IN_*, *R_OUT_* and *X_OUT_* are the real and imaginary parts of *Z_IN_* and *Z_OUT_*, respectively. Figure 3a–d show the impedance matching network topology between the excitation and demodulation module and the inputs of the EUE channel. Figure 3e–h show the impedance matching network topology of the EUE channel outputs with the receiver and sensor module.

When *R_S_*, *R_IN_* and *X_IN_* satisfy the two cases, as shown in Figure 3a,b, the inductance and capacitance values in their impedance matching networks are calculated as follows:(4)$$\left\{ \begin{array}{l} C_{1}=\frac{1}{2\pi f(\sqrt{R_{IN}(R_{S}-R_{IN})}+X_{IN})}\\ L_{1}=\frac{R_{S}}{2\pi f}\sqrt{\frac{R_{IN}}{R_{S}-R_{IN}}}\\ L_{2}=\frac{-(\sqrt{R_{IN}(R_{S}-R_{IN})}+X_{IN})}{2\pi f} \end{array}\right.$$

When *R_S_*, *R_IN_* and *X_IN_* satisfy the two cases, as shown in Figure 3c,d, the inductance and capacitance values in their impedance matching networks are calculated as follows:(5)$$\left\{ \begin{array}{l} C_{1}=\frac{1}{2\pi f(\sqrt{R_{S}(R_{IN}-R_{S})})}\\ C_{2}=\frac{1}{2\pi fX_{IN}}\\ L_{1}=\frac{R_{IN}}{2\pi f}\sqrt{\frac{R_{S}}{R_{IN}-R_{S}}}\\ L_{2}=\frac{-X_{IN}}{2\pi f} \end{array}\right.$$

When *R_L_*, *R_OUT_*, and *X_OUT_* satisfy the two cases, as shown in Figure 3e,f, the inductance and capacitance values in their impedance matching networks are calculated as follows:(6)$$\left\{ \begin{array}{l} C_{3}=\frac{1}{2\pi f(\sqrt{R_{L}(R_{OUT}-R_{L})})}\\ C_{4}=\frac{1}{2\pi fX_{OUT}}\\ L_{3}=\frac{R_{OUT}}{2\pi f}\sqrt{\frac{R_{L}}{R_{OUT}-R_{L}}}\\ L_{4}=\frac{-X_{OUT}}{2\pi f} \end{array}\right.$$

When *R_L_*, *R_OUT_*, and *X_OUT_* satisfy the two cases, as shown in Figure 3g,h, the inductance and capacitance values in their impedance matching networks are calculated as follows:(7)$$\left\{ \begin{array}{l} C_{3}=\frac{1}{2\pi f(\sqrt{R_{OUT}(R_{L}-R_{OUT})}+X_{OUT})}\\ L_{3}=\frac{R_{L}}{2\pi f}\sqrt{\frac{R_{OUT}}{R_{L}-R_{OUT}}}\\ L_{4}=\frac{-(\sqrt{R_{OUT}(R_{L}-R_{OUT})}+X_{OUT})}{2\pi f} \end{array}\right.$$

### 3.2. Impedance Modulation

Impedance modulation is achieved by two MOSFETs controlling the connection and disconnection of the modulation resistor *R_M_*. When *R_M_* is disconnected, both ends of the MEUE channel are in a simultaneous conjugate impedance matching state, and the energy transfer efficiency is maximized. When *R_M_* is connected, the simultaneous conjugate matching state of the MEUE channel is broken, and the energy transfer efficiency is reduced. This process changes the amplitude of input voltage and output voltage of the MEUE channel. When the amplitude is high, the binary data “1” is transmitted, and when the amplitude is low, the binary data “0” is transmitted. During the transmitting of the binary data, the energy transfer process is not interrupted. Therefore, the coupling of ultrasonic energy transfer and data communication is realized.

According to the equivalent circuit model shown in Figure 2, the EUE channel is cascaded with an impedance matching network to form the MEUE channel. To investigate the effects of different resistance values of the *R_M_* on the voltage at the ends of the MEUE channel, the system energy transfer efficiency and the modulation coefficient, the Z-parameters of the MEUE channel, need to be obtained first.

As shown in Figure 2, the T-parameter matrices of the impedance matching network at the input and output are *TN1* and *TN3*, and the T-parameter matrix of the EUE channel is *TN2*. Since the *T*-parameters and *Z*-parameters can be converted to each other, the *Z*-parameters of the MEUE channel are shown in Equation (8) [23]:(8)$$\left\{ \begin{array}{l} \left[\begin{array}{cc} TN_{11} & TN_{12}\\ TN_{21} & TN_{22} \end{array}\right]=TN_{1}TN_{2}TN_{3}\\ \left[\begin{array}{cc} ZN_{11} & ZN_{12}\\ ZN_{21} & ZN_{22} \end{array}\right]=\left[\begin{array}{cc} \frac{TN_{11}}{TN_{21}} & \frac{TN_{11}TN_{22}-TN_{12}TN_{21}}{TN_{21}}\\ \frac{1}{TN_{21}} & \frac{TN_{22}}{TN_{21}} \end{array}\right] \end{array}\right.$$

Based on the *Z*-parameters of the MEUE channel and the circuit model shown in Figure 2, the amplitude expressions for *V*_1_ and *V*_2_ when the MOSFETs are on or off can be derived. When the MOSFETs are turned off, the amplitudes of *V*_1_ and *V*_2_ are shown in Equation (9), where |*ZN*| is the determinant of the *Z*-parameter matrix of the MEUE channel and *ZN_INOFF_* is the input impedance of the MEUE channel:(9)$$\left\{ \begin{array}{l} V_{1OFF}=\frac{ZN_{INOFF}U_{S}}{ZN_{INOFF}+R_{S}}\\ V_{2OFF}=\frac{R_{L}ZN_{21}V_{1OFF}}{\left(|ZN|+ZN_{11}R_{L}\right)}\\ ZN_{INOFF}=\frac{|ZN|+ZN_{11}R_{L}}{R_{L}+ZN_{22}} \end{array}\right.$$

When the MOSFETs are turned on, the input impedance of the MEUE channel becomes *ZN_INON_*, and the amplitudes of *V*_1_ and *V*_2_ are shown in Equation (10). Here, the MOSFETs are regarded as an ideal element with zero internal resistance at conduction.
(10)$$\left\{ \begin{array}{l} V_{1ON}=\frac{ZN_{INON}U_{S}}{ZN_{INON}+R_{S}}\\ V_{2ON}=\frac{R_{M}R_{L}ZN_{21}V_{1ON}}{|ZN|(R_{M}+R_{L})+ZN_{11}R_{L}R_{M}}\\ ZN_{INON}=\frac{|ZN|(R_{M}+R_{L})+ZN_{11}R_{L}R_{M}}{ZN_{22}(R_{M}+R_{L})+R_{M}R_{L}} \end{array}\right.$$

Based on the above process, the amplitudes of *V*_1_ and *V*_2_ change when the MOSFETs control *R_M_* to disconnect or connect, i.e., amplitude modulation is achieved. To evaluate the effect of different modulation resistors *R_M_* on the energy transfer and data communication of the system, the energy transfer efficiency *η_m_* and the amplitude modulation coefficient *Ma* are introduced to measure the energy transfer and data communication performance of the system. The formulas for *Ma* and *η_m_* are shown below:(11)$$\left\{ \begin{array}{l} Ma=|\frac{V_{1ON}-V_{1OFF}}{V_{1ON}+V_{1OFF}}|\\ \eta_{m}=\frac{1}{2}|\frac{V_{2OFF}^{2}ZN_{INOFF}}{V_{1OFF}^{2}R_{L}}+\frac{V_{2ON}^{2}ZN_{INON}(R_{L}+R_{M})}{V_{1ON}^{2}R_{L}R_{M}}| \end{array}\right.$$

The modulation coefficient *Ma* is the ratio of the maximum amplitude to the minimum amplitude of *V*_1_. The smaller *Ma* is, the lower the quality of communication and the more difficult it is to demodulate the sensor data in the carrier signal. *Ma* and *η_m_* are affected by *R_M_*, so the coupling of energy transfer and data communication can be further optimized by choosing the suitable resistance value of *R_M_*.

## 4. Results

To verify the validity of the equivalent circuit model shown in Figure 2, we constructed a test system, as shown in Figure 4. SMC is a square 304 stainless steel enclosure. The IPZT and OPZT were coaxially secured on the 11 mm thick side to form the EUE channel. The EUE channel was connected to the impedance matching network to form the MEUE channel. The external excitation and demodulation module was powered by a DC power supply. The signal generated by the AD9833 chip was amplified by the power amplifier and became the carrier signal driving the MEUE channel. The receiver and sensor module converts the AC output from the MEUE channel into a stable DC and controls the disconnection or connection of the *R_M_* through an impedance modulation circuit to achieve amplitude modulation of the carrier signal. An oscilloscope (DSOX3024T) is used to test the amplitude of the input and output voltages of the MEUE channel. An LCR digital bridge (IM3536) is used to test the input impedance of the MEUE channel.

### 4.1. System Testing

The S-parameter scan test for the EUE channel using a vector network analyzer is shown in Figure 4 (ROHDE & SCHWARZ ZNB20, Taiyuan, China). The characteristic impedance *Z*_0_ of the ROHDE & SCHWARZ ZNB20 was 50 Ω. By substituting the scanning S-parameters and *Z*_0_ into Equations (1)–(3) and (12), the S-parameters of the MEUE channel can be determined for different frequencies.
(12){[SN11SN12SN21SN22]=F[Z11−ZINZ12Z21Z22−ZOUT][Z11+ZIN*Z12Z21Z11+ZOUT*]−1F−1F=[2Re{ZIN*}−1002Re{ZOUT*}−1]

The energy transfer efficiency can be expressed by the positive transfer coefficient (*S*_21_) in the *S*-parameter, as shown in Equation (13):(13)$$\eta =\max\{|S_{21}{|}^{2}\}\times 100\%$$

The excitation and demodulation module acts as a signal source, while the receiver and sensor module plays the role of a load. The energy conversion efficiency of the system can be measured by the energy transfer efficiency from the source to the load. To verify the improvement in the synchronous conjugate impedance matching technique on the energy transfer efficiency of the system, comparative experiments were conducted. Figure 5 shows the comparison of energy transfer efficiency for transferring energy from source to load without impedance matching and with simultaneous conjugate impedance matching. Without impedance matching, the source-to-load energy transfer efficiency is a maximum of 53% at a frequency of 1.048 MHz. After simultaneous conjugate impedance matching, the energy transfer efficiency is significantly improved in the 0.5 to 1.5 MHz band, and the best performance is achieved at a frequency of 1.045 MHz with an energy transfer efficiency of 64%. Therefore, the carrier signal frequency is set to 1.045 MHz to ensure the most efficient energy transfer in the system.

When the frequency was 1.045 MHz, the *S*-parameters of the EUE channel are shown in Table 1.

The Z-parameters of the EUE channel are shown in Table 2 for when the frequency *f* was 1.045 MHz, t. The input impedance *Z_IN_* and output impedance *Z_OUT_* for simultaneous conjugate impedance matching of the EUE channel can be obtained by substituting the *Z*-parameters into Equations (2) and (3) in Section 3.1.

The MOSFETs in the receiver and sensor module are switched off by default with an equivalent load impedance *R_L_* of 200 Ω. The internal impedance *R_S_* of the excitation and demodulation module was 50 Ω and the frequency *f* was 1.045 MHz. The EUE channel input impedance *Z_IN_* and output impedance *Z_OUT_* are shown in Table 2, and were 32.6−j37.1 Ω and 40.3−j35.4 Ω, respectively. Based on these parameters, the topology of the impedance matching network in the test system was determined, as shown in Figure 3b,g in Section 3.1. The calculation procedure is described in Equations (4) and (7) in Section 3.1, and the results are presented in Table 3.

By substituting the parameters in Table 2 and Table 3 into Equation (8) in Section 3.1, the Z-parameters of the MEUE channel with a frequency *f* of 1.045 MHz can be obtained, and the specific values are shown in Table 4.

When the frequency *f* was 1.045 MHz, the equivalent circuits of the system without impedance matching and with simultaneous conjugate impedance matching are shown in Figure 6a,b according to the Z parameters in Table 2 and Table 4. The equivalent circuit diagram of the test system shown in Figure 4 is shown in Figure 6b. Since *R_S_* was 50 Ω and *R_L_* was 200 Ω, theoretically *ZN_INOFF_* should be 50 Ω and *ZN_OUTOFF_* should be 200 Ω when *R_M_* is disconnected. In the actual test, *ZN_INOFF_* and *ZN_OUTOFF_* were measured as 51∠5° Ω and 198∠9° Ω, respectively, by the LCR digital bridge (IM3536) at a frequency of 1.045 MHz. The deviation between theory and practice was small, and conjugate impedance matching was achieved at both ends of the MEUE channel with both the source and the load.

The *U_S_* amplitude of the excitation and demodulation module was set to 20 V, with a frequency *f* of 1.045 MHz. When the *R_M_* was disconnected, we used an oscilloscope (DSOX3024T) to measure the voltage waveforms at both ends of the EUE channel in Figure 6a and the MEUE channel in Figure 6b, as shown in Figure 7a,b. The voltage amplitudes of the EUE channel were 9.0 V and 9.8 V, while the voltage amplitudes of the MEUE channel were 10.2 V and 15.8 V. Active power can be calculated by substituting the measured voltage value into Equation (14):(14)$$P_{AP}=\frac{(V_{OUT}/\sqrt{2}{)}^{2}}{R_{L}}$$

The active power output from the EUE channel in Figure 5a and the MEUE channel in Figure 5b were 240 mW and 624 mW, respectively. If we reduce the amplitude of the *U_S_*, the test system, connected as shown in Figure 5a, will not work properly, which means that the minimum active power required by the receiver and sensor modules was 240 mW. After connecting the *R_M_*, Figure 7c–e shows the effect of different *R_M_* on the voltages *V*_1_, *V*_2_, and *ZN_INON_* on the MEUE channel in Figure 5b, with the *R_M_* gradually increasing from 0 to 1 kΩ. The solid line is the theoretical result according to Equation (10). The dotted line is the test result of DSOX3024T and IM3536. The theoretical and measured coincide with each other, proving the correctness of the theoretical model. When the *R_M_* was set to 300 Ω and the data communication rate was 50 kbps, the test waveforms of *V*_1_ and *V*_2_ are shown in Figure 7f,g. When transmitting binary data “1”, the MOSFETs control *R_M_* to be disconnected, and the amplitudes of *V*_1_ and *V*_2_ were 10.2 V and 15.8 V, respectively; when transmitting binary data “0”, the MOSFETs control *R_M_* to be connected, and the amplitudes of *V*_1_ and *V*_2_ were 8.45 V and 11.5 V. According to Figure 7e, the tested value of *ZN_INON_* was about 37.1 Ω. Substituting into Equation (11) to calculate, the energy transfer efficiency *η_m_* was 60% and the modulation coefficient *Ma* was 0.1.

The effects of different modulation resistances *R_M_* on *η_m_* and ma are shown in Figure 7h,i. When *R_M_* is gradually increased from 0 to 1 kΩ, *Ma* decreases with the increase in *R_M_* and *η_m_* increases with the increase in *R_M_*. When the resistance of *R_M_* was greater than 300 Ω, the modulation coefficient *Ma* was less than 0.1, the voltage amplitude of *V*_1_ changed little, and the communication quality was poor. Therefore, the modulation resistance of the system was fixed at 300 Ω. According to the above measurements, the energy transfer efficiency of the system was 60% and the data communication rate was 50 kbps, which was higher than other studies in terms of the energy transfer efficiency and communication rate [12,13,15,19,23,24,28].

### 4.2. Practical Application

To verify the feasibility of the proposed system for the ultrasonic CM of the SMC, the integrated system was installed on an SMC, as shown in Figure 8. This simulates the process of CM in actual engine compartments.

The IPZT and OPZT were fixed to either side of the bottom of the SMC, which is a cylindrical stainless steel with a 13 mm thick base. The receiver and sensor module was mounted inside the SMC and was used to monitor temperature and humidity and return sensor data. An external excitation and demodulation module powers the internal receiver and sensor module and demodulates the returned sensor data. In addition, a commercial standard manometer was installed in SMC. The manometer showed that the pressure inside SMC was 15.9 kPa at this point, indicating that the SMC was completely sealed.

The receiver and sensor module within SMC converts the monitored temperature and humidity data into a fixed-format binary array. The binary array has a total of 40 bits, with a start bit of 0011, a cut-off bit of 1100, and 24 bits in the middle for the data bits, of which the high 16 bits represent the temperature data and the low 8 bits represent the humidity data within. The voltage waveform of *V*_1_ when the receiver and sensor module achieve amplitude modulation of the carrier signal (*V*_1_), by means of an impedance modulator at a rate of 50 kbps, is shown in Figure 9a. The temperature and humidity data are loaded in the envelope signal. After envelope detection, filtering, amplification and voltage comparison, the envelope signal is converted into a binary analogue signal, as shown in Figure 9b. When a binary number “0” is returned, the output voltage of the voltage comparator is low with a hold time of 20 μs. When a binary number “1” is returned, the output voltage of the voltage comparator is high, with a hold time of 20 μs. The default output of the voltage comparator is high, and when it outputs a low level for the first time, it means that data communication starts. The MCU samples the analogue signal output from the voltage comparator with a sampling frequency of 500 kHz, and finally converts it into a digital signal after the MCU judgement and outputs it from the IO of the MCU, as shown in Figure 9c. Figure 9c shows that the temperature and humidity data at this point were 29.9 °C and 39%, respectively. The standard thermo-hygrometer shows that the temperature inside SMC was 30.2 °C and the humidity was 40%, which is consistent with the demodulation results.

## 5. Conclusions

This paper demonstrated a wireless and nondestructive monitoring system utilizing ultrasound, integrating the energy transfer and data communication functions for the wireless condition of SMC. The system utilizes simultaneous conjugate impedance matching and impedance modulation techniques to achieve the effective coupling of energy transfer and data communication in the same channel. An equivalent circuit model of the system was designed, and the impedance matching network and impedance modulation process were analyzed and optimized in detail. The experimental results show that the proposed system was consistent with the theoretical modeling results. The system has the advantages of high integration, high efficiency, and good stability, and can be used for the multi-parameter environmental monitoring of sealed metal cavities. The technique is of great significance for the condition monitoring of specific devices that cannot withstand physical penetration, and has promising applications in the aerospace, marine, industrial, and nuclear energy fields. The existing system is still only capable of internal-to-external data communication with a maximum communication rate of 50 kbps. In future work, the system can combine phase shift modulation, frequency shift modulation and amplitude modulation to achieve bi-directional data communication in the system and increase the communication rate of the system. The designed internal receiver and sensor module can be further miniaturized by MEMS technology to suit more applications.

## Figures and Tables

**Figure 1 micromachines-15-00048-f001:**
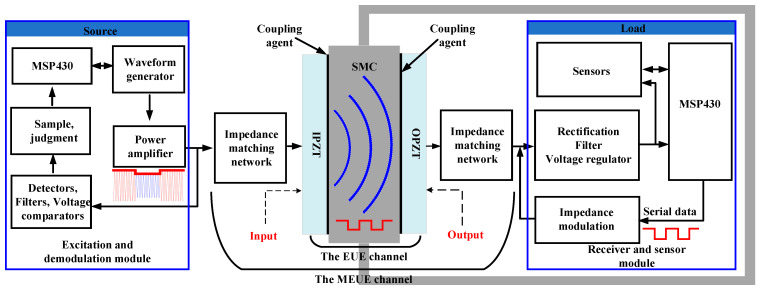
The block diagram of the ultrasonic CM system.

**Figure 2 micromachines-15-00048-f002:**
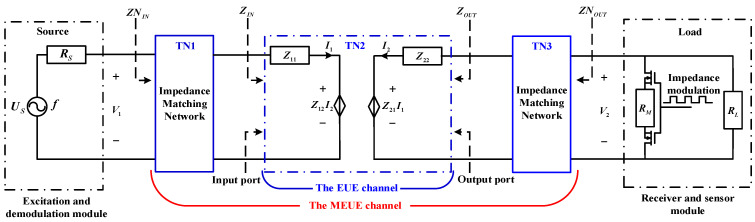
System equivalent circuit model.

**Figure 3 micromachines-15-00048-f003:**
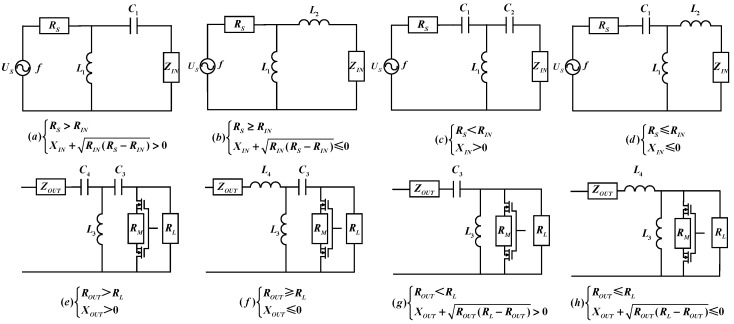
Impedance matching topology structure. (**a**) Source L-type impedance matching network. (**b**) Source L-type impedance matching network. (**c**) Source T-type impedance matching network. (**d**) Source T-type impedance matching network. (**e**) Load T-type impedance matching network. (**f**) Load T-type impedance matching network. (**g**) Load L-type impedance matching network. (**h**) Load L-type impedance matching network.

**Figure 4 micromachines-15-00048-f004:**
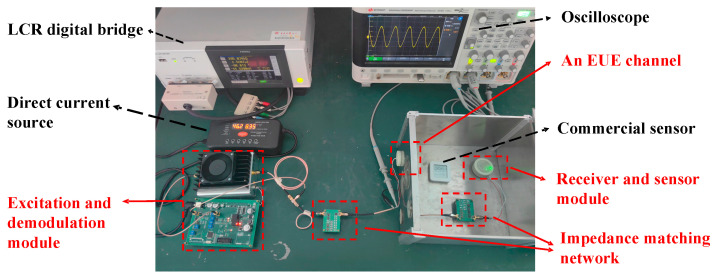
The test system penetrated an 11 mm thick steel wall.

**Figure 5 micromachines-15-00048-f005:**
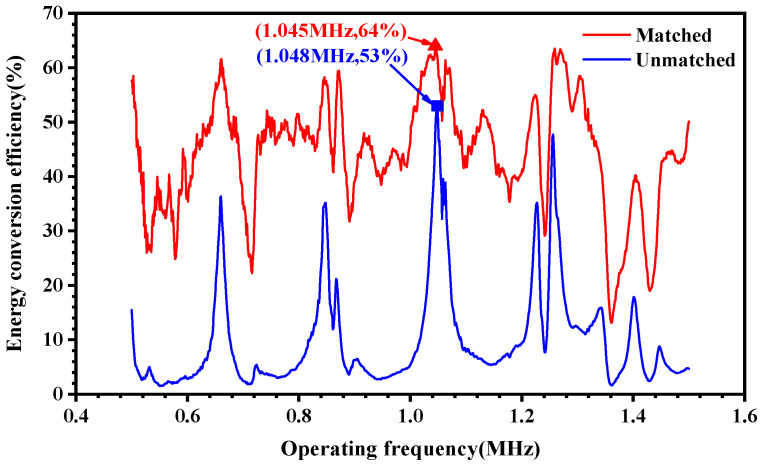
Comparison of the efficiency of energy transfer from the source to the load without impedance matching and with simultaneous conjugate impedance matching.

**Figure 6 micromachines-15-00048-f006:**
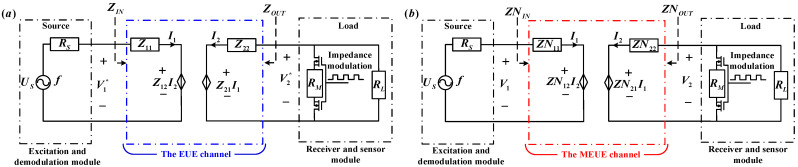
The equivalent circuit models for two test systems. (**a**) The equivalent circuit diagram of the system without impedance matching. (**b**) The equivalent circuit diagram of the system after simultaneous conjugate impedance matching.

**Figure 7 micromachines-15-00048-f007:**
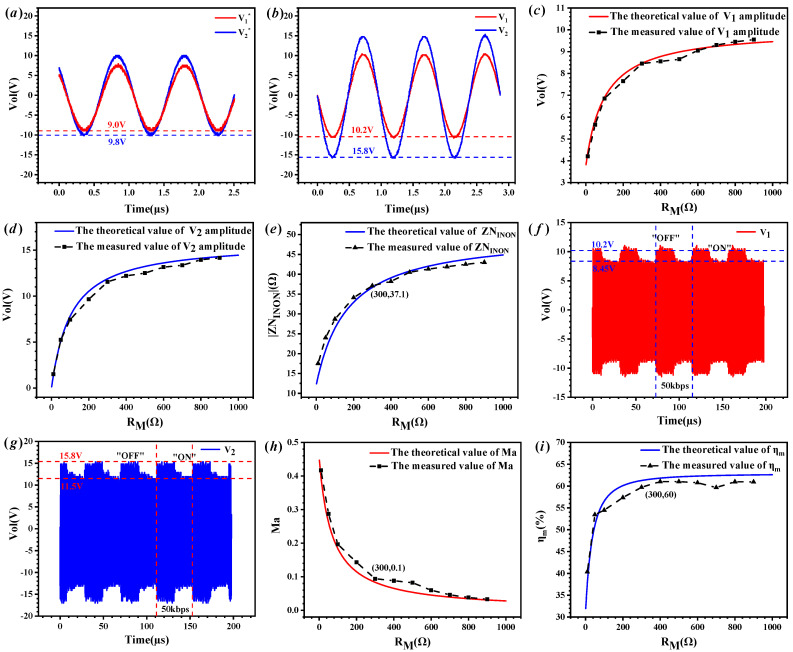
Energy transfer and data communication processes of the system. (**a**) Voltage profile on the EUE channel in Figure 5a when the *R_M_* is disconnected. (**b**) Voltage profile on the MEUE channel in Figure 5b when the *R_M_* is disconnected. (**c**) The effect of different *R_M_* on the voltage *V*_1_ of the MEUE channel when the *R_M_* is connected. (**d**) The effect of different *R_M_* on the voltage *V*_2_ of the MEUE channel when the *R_M_* is connected. (**e**) The effect of different *R_M_* on the |*ZN_INON_*| of the MEUE channel when the *R_M_* is connected. (**f**) The waveform of *V*_1_ when *R_M_* is 300 Ω and the system communication rate is 50 kbps. (**g**) The waveform of *V*_2_ when *R_M_* is 300 Ω and the system communication rate is 50 kbps. (**h**) The effect of different *R_M_* on *Ma*. (**i**) The effect of different *R_M_* on *η_m_*.

**Figure 8 micromachines-15-00048-f008:**
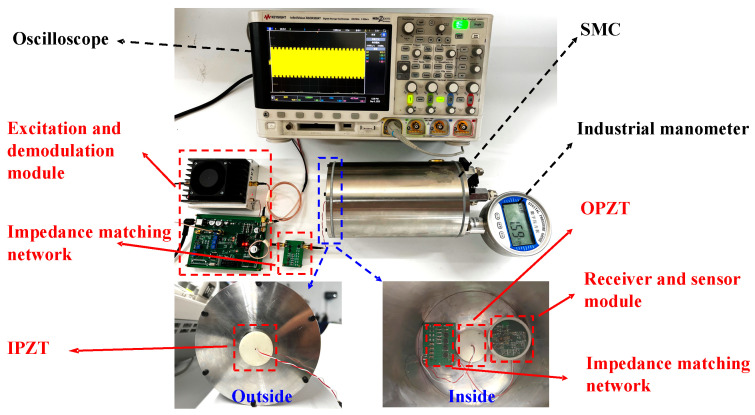
Experimental setup for the ultrasonic CM of the SMC.

**Figure 9 micromachines-15-00048-f009:**
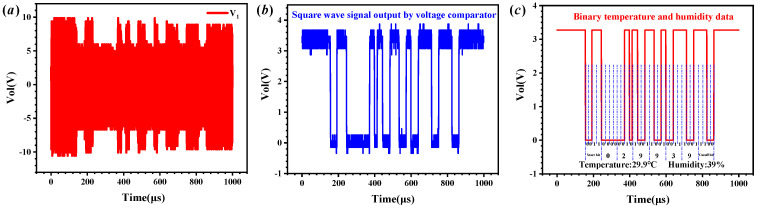
Demodulation process of temperature and humidity data: (**a**) the voltage waveform of *V*_1_; (**b**) the voltage waveform output from the voltage comparator; and (**c**) demodulated temperature and humidity data.

**Table 1 micromachines-15-00048-t001:** The S-parameters of the EUE channel.

*S* _11_	*S* _12_	*S* _21_	*S* _22_
−0.184−j0.351	0.408−j0.569	0.409−j0.567	0.201−j0.274

**Table 2 micromachines-15-00048-t002:** The Z-parameters, the input impedance *Z_IN_* and output impedance *Z_OUT_* of the EUE channel.

*Z* _11_	*Z* _12_	*Z* _21_	*Z* _22_	*Z_IN_*	*Z_OUT_*
7.7−j28.6 Ω	−2.3−j36.5 Ω	−2.2−j36.5 Ω	9.5−j24.9 Ω	32.6−j37.1 Ω	40.3−j35.4 Ω

**Table 3 micromachines-15-00048-t003:** Impedance matching network parameters.

*L* _1_	*L* _2_	*L* _3_	*C* _3_
10.4 μH	2.0 μH	15.3 μH	3.4 nF

**Table 4 micromachines-15-00048-t004:** The Z-parameters of the MEUE channel.

*ZN* _11_	*ZN* _12_	*ZN* _21_	*ZN* _22_
190.5−j83.3 Ω	368.8−j171.6 Ω	369.2−j170.6 Ω	759.1−j331.6 Ω

## Data Availability

The data presented in this study are available on request from the corresponding author.

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
