# Peer review of "Optimized Design of an Ultrasonic-Based High-Efficiency Wireless Passive Monitoring System for Sealed Metal Compartments"

_micromachines, 2023, doi:10.3390/mi15010048_

Round 1

Reviewer 1 Report (Previous Reviewer 2)

Comments and Suggestions for Authors

A major revision was done and the authors looked thoroughly at the reviewers point. This is highly appreciated and is favourably received.

However still the details on descriptions are insufficient, e.g. you have to specify which PZTs were used (material specs from manufacturer, outer dimensions, orientation of polarization and electrodes; or provide details so that other scientist are able to replicate your experiments, e.g. specification like "P-88x.x5 Encapsulated PICMA from PI Ceramic GmbH")

Please provide a reference where to find Thevenin's theorem.

Having read the paper more than ones I was surprised that the full article does not provide any number about the power. Please answer how much power (Watt) is transferred over the EUE and over the MEUE and how high the power demand of the load (cp. Figure 1) was.

My suggestion is that you link the mathematical description in line 207ff to the figures 1 and 2, so that it is clear which parts of the setup depicted in figure 1 and 2 actually are described by TN1, TN2 and TN3.

In figure 5: Does the energy conversion efficiency of the matched system take into account the power demand of the impedance matching network? This will lower the efficiency.

I fully understand how you can increase the efficiency by impedance matching. However, I do not fully understand how the efficiency of the MEUE can be higher than of the EUE. If by accident the EUE is impedance matched it will transfer a certain power. If I then add matching electronics they will need power and thus the MEUE must have lower efficiency than the EUE which is accidently matched. My suggestion is that the authors are super clear in their definition of the efficiency and perhaps even look at it in two perspectives: One looking at the efficiency from source to load and one that looks only at the EUE.

The EUE in figure 1 does not include the impedance matching network. However, in figure 2 the EUE channel does include the matching network. Please to clarify and check your whole paper for further inconsistencies. Figure 3 indicates that figure 2 is wrong and figure 1 is right.

I also found no consistent way how to merge the figures from figure 3 with figure 2 and how to get a consistent view with figure 6.

Overall more details have to be added and the description has be made more consistent and clear, i.e. do cross reference between the different parts of the paper and be specific on how to put the pieces together.

Comments on the Quality of English Language

line 10 has a "the" too much

line 12 "ultrasonic" is used when "ultrasound" should be used. It seems that the difference between an adjective and a noun is not clear. This error was done several times. e.g. line 55

there has to be not comma after "e.g.", "i.e." ... ; examples where this went wrong e.g. line 83, line 223

Author Response

We would like to thank the referees for your efforts in reviewing our manuscript titled "Optimized design of an ultrasonic-based high-efficiency wireless passive monitoring system for sealed metal compartments", and providing many helpful comments and suggestions, which will all prove invaluable in the revision and improvement of our paper, as well as in guiding our research in the future. 

We have studied your comments point by point and revised the manuscript accordingly. The response to the referees' comments and point-by-point changes are in the document "Response Letter to the Referees & Change List for Referees", and all authors have approved the response letter and the revised version of the manuscript. 

Reviewer 2 Report (Previous Reviewer 1)

Comments and Suggestions for Authors

Most of my questions have been answered clearly, and the article has reached a publishable state

Author Response

We would like to thank the referee for your efforts in reviewing our manuscript entitled "Optimized design of an ultrasonic-based high-efficiency wireless passive monitoring system for sealed metal compartments" and feel very honoured to have your endorsement of this work.

Round 2

Reviewer 1 Report (Previous Reviewer 2)

Comments and Suggestions for Authors

Dear authors,

thank you for revising your paper in a very detailed way.

My last point is that the quality of the figures 4 and 8 should be checked. They look awfull when I look at them in the browser and are ok in the pdf reader.

This manuscript is a resubmission of an earlier submission. The following is a list of the peer review reports and author responses from that submission.

Round 1

Reviewer 1 Report

Comments and Suggestions for Authors

A miniaturized and efficient ultrasonic condition monitoring system for sealed metal containers is proposed in this paper. The system utilizes ultrasonic waves to power and communicate with internal sensors, enabling non-destructive monitoring of the container's condition. The system achieves efficient wireless power transfer and effective data communication, with a high energy transfer efficiency of 60% and a communication rate of 50 kbps. It also enables temperature and humidity monitoring inside the container, providing a practical and effective solution for condition monitoring in various fields.

The article proposes a relatively new method that has been validated and has a certain degree of feasibility and novelty. However, some issues need to be comprehensively corrected before the article can be published

1.Elaborate more on the theoretical basis and fundamental principles behind the proposed optimal design method.
2.Provide more details on the equivalent circuit modeling process and the significance of obtaining Z-parameters.
3.Explain the rationale for choosing the specific operating frequency of 1.045 MHz.
4.Expand the description of the system setup and installation process for the prototype.
5.Include more specifics on the testing process, like sampling frequency, data processing methods, etc.
6.Provide quantitative data analysis and insight into how the results align with theoretical values.
7.Discuss how the experimental results could be further improved and what can be done in future work.

Reviewer 2 Report

Comments and Suggestions for Authors

Dear authors,

I found the overall topic of the paper „A miniaturized and efficient ultrasonic condition monitoring system of the sealed metal container” very interesting. But the trouble started when I tried to find some aspects of the title in the paper itself.

I really did search through the whole paper and I did not find the aspect of miniaturization being addressed anywhere besides the motivating introduction. Since miniaturization aspects are not addressed at all I would request that miniaturized is eliminated from the title or miniaturization is thoroughly being addressed in the main part of the paper.

More striking I found the same problem with ultrasonic. Ultrasound is mentioned well also beyond the introduction and it is used in the experiments. However, Figure 2 clearly shows that no ultrasonic aspects are used within the modelling. Figure 1 included the coupling agents, but their effect is somehow glossed over in some coupling constants that are no discussed within the paper. (Where is the SMC in figure 2 and how are changes in the SMC impact on the equivalent circuit.) The unloving dealing with acoustics is also reflect in the positioning of the receiving module being shown in figure 4. This is fully different as in figure 1. The location is really surprising because it is in quite some distance and in 90° to the outside unit. However, it might be that the actual geometry is different, which stresses how easily one can misunderstand the presentation of the experimental setup.

I also have trouble with the experimental setup. Virtually no information besides the system equivalent circuit and that piezo electric elements are used are provided. On this basis is will be impossible by any person in the research community to prove or disprove the research finding. I request a much more detailed presentation that allows the replication of the work, e.g. exact specification on the piezo electrics used, including the backing material, fronting material, …; exact specification of the steel, exact specification of the electronics, sensors, …. being used.

I also tried to follow a bit more the simulation part. But I again had to learn that the paper is not written in a way that the results can be easily be verified or falsified. Equation 2 provides the equations how to get the Z values, if the scattering matrix S values are given. However, the authors do cite only reference 26 and therefore explains where these equations come from and do not provide any information where and how to get the S values from. Hence, one unknown is replaced by another unknow. The significance of \alpha and \Delta is not explained. Next matching topology structures are introduced. To be able to determine which one to use, one has to know R_S and R_IN. R_S, R_M, R_L can be found in Figure 2. How one gets R_IN etc. is left to the reader. Also, Figure 2 and Figure 3 are not trivially matched. I tried to figure out how Z11 and Z12 impact on ZIN … and found not figure that explains this.

From my point of view the whole theory part including is equations has to be rewritten so that it is possible to follow and so that it is clear where one can get which value from the experiment for the simulation and the other way around. The acoustic energy transfer will change, if piezo properties are changed. Where is this reflected in the model (beyond the pure electric properties)? Chances in the coupling layer, backing, SMC, orientation of the transducers, … will all impact on the energy transfer. Right now, I do not see where this is reflected. If you move from rolled steel to casted iron and the acoustic attenuation with rise without necessarily impacting on the impedances of the piezoelectric transducers. This shows, that the model as set up is incomplete.

From the paper there is no good explanation why the limiting efficiency is about 60%. I would expect that the authors explain with their paper where this limit comes from and how this limiting value can the altered.